# Coulomb Contribution to Shockley–Read–Hall Recombination

**DOI:** 10.3390/ma17184581

**Published:** 2024-09-18

**Authors:** Konrad Sakowski, Pawel Strak, Pawel Kempisty, Jacek Piechota, Izabella Grzegory, Piotr Perlin, Eva Monroy, Agata Kaminska, Stanislaw Krukowski

**Affiliations:** 1Institute of Applied Mathematics and Mechanics, University of Warsaw, 02-097 Warsaw, Poland; konrad@unipress.waw.pl; 2Institute of High Pressure Physics, Polish Academy of Sciences, 01-142 Warsaw, Poland; strak@unipress.waw.pl (P.S.); jpa@unipress.waw.pl (J.P.); izabella@unipress.waw.pl (I.G.); piotr@unipress.waw.pl (P.P.); kaminska@ifpan.edu.pl (A.K.); 3Quantum Photonics, Electronics and Engineering Laboratory (PHELIQS), CEA-Grenoble, 38000 Grenoble, France; eva.monroy@cea.fr; 4Institute of Physics, Polish Academy of Sciences, 02-668 Warsaw, Poland; 5Faculty of Mathematics and Natural Sciences, School of Exact Sciences, Cardinal Stefan Wyszynski University, 01-815 Warsaw, Poland

**Keywords:** SRH recombination, semiconductors, photoluminescence

## Abstract

A nonradiative recombination channel is proposed, which does not vanish at low temperatures. Defect-mediated nonradiative recombination, known as Shockley–Read–Hall (SRH) recombination, is reformulated to accommodate Coulomb attraction between the charged deep defect and the approaching free carrier. It is demonstrated that this effect may cause a considerable increase in the carrier velocity approaching the recombination center. The effect considerably increases the carrier capture rates. It is demonstrated that, in a typical semiconductor device or semiconductor medium, the SRH recombination rate at low temperatures is much higher and cannot be neglected. This effect renders invalid the standard procedure of estimating the radiative recombination rate by measuring the light output in cryogenic temperatures, as a significant nonradiative recombination channel is still present. We also show that SRH is more effective in the case of low-doped semiconductors, as effective screening by mobile carrier density could reduce the effect.

## 1. Introduction

Shockley–Read–Hall (SRH) recombination is a nonradiative process that involves a defect and an electron-hole pair. SRH is one of the most important detrimental processes for optoelectronic devices, having a negative impact on the performance of all semiconductor light emitters, including laser diodes (LDs), light-emitting diodes (LEDs), superluminescent light-emitting diodes (SLEDs), and electroluminescent lamps [1,2]. Therefore, investigations of the basic properties of SRH recombination are of great interest from the fundamental point of view and also for applications. The basic features of the process were described relatively early, independently by Shockley and Read [1] and by Hall [3]. Since then, the primary role of the SRH process in the description of the energy dissipation and emission processes has remained unchanged [4,5,6].

The recent advent of nitride optoelectronic confirmed the important role of SRH recombination. However, the optical properties of these materials present a higher degree of complexity due to the initial high density of defects, alloy inhomogeneity, polarization, complex Auger processes, and strong electron–phonon interactions. Hence, the peculiarities of the SRH process were difficult to assess, and the role of SRH recombination in the overall performance is still under discussion [7,8,9,10]. It is widely recognized that, in InGaN LEDs, carrier localization in In-rich areas prevents migration to dislocation lines, which usually serve as effective nonradiative SRH recombination centers in semiconductors [7,8,9]. Such a positive insulation effect is not observed in AlGaN-based devices, which leads to a lower emission efficiency in comparison to their InGaN-based counterparts [11]. Recent application of Mg–Si codoping opens new avenues toward higher efficiency deep ultraviolet LEDs that may contribute significantly to overcoming important obstacles in the development of these devices [12].

The role of SRH recombination is universally recognized despite its relatively simple model. It is assumed that a deep defect level, i.e., the state located deep in the bandgap, is occupied by the majority carrier. As the energy of the state is much below the conduction band minimum of the crystal host and also much below the Fermi energy level, we can assume that the occupation of the deep localized state is close to complete. In the SRH process, the minority carrier is “captured” by the defect to be recombined. This is the slowest rate-controlling step; therefore, SRH is effectively a mono-molecular process with a recombination rate that is proportional to the density of minority carriers. The net result is the thermalization of the excess energy in the form of lattice vibrations, i.e., phonons that effectively heat the sample. Thus, the effect is doubly detrimental, first by the loss of energy and second by the increase in temperature, which decreases device efficiency and can ultimately lead to the degradation of the device.

The SRH recombination model, devised by Shockley, is based on a classical description of the minority carrier collision with the deep defect occupied by the majority carrier [1,3,4]. The recombination rate is calculated from the average capture rate of the minority carrier by the defect. This collision cross-section depends on the defect type (point or linear). Regardless of the type of defect, a shared feature of both types is that the size of the deep defect falls within the order of a lattice constant *a*. The cross-section is calibrated to experimental data, sometimes supplemented by the capture probability factor. This description has facilitated a basic modeling of optoelectronic devices, albeit with reduced prediction accuracy.

Due to the nature of SRH recombination, it is assumed that this recombination channel is suppressed in low temperatures. To study the properties of radiative recombination, the standard approach is to study samples in cryogenic temperatures, as it is generally believed that then only the radiative recombination channel would be significant. In this paper, we demonstrate that this is not the case, at least if the majority carrier concentrations are sufficiently low. If this is the case, the charge screening length is high, the Coulomb interaction between charged deep defects and minority charge carriers is high, and the nonradiative recombination rate is significantly increased. The effect is particularly important for the development of luminescent semiconductor devices.

SRH recombination rates can be extracted from time-resolved photoluminescence (TRPL) measurements. However, directly determining the SRH rate from TRPL data is challenging because the effect is entangled with the other processes, including radiative or Auger recombination [13]. Recently, we proposed a new method for the analysis of TRPL data, which allows disentangling mono-, bi-, and tri-molecular contributions to the overall photoluminescence decay [14,15]. However, the obtained results do not support the simple attribution of mono-molecular processes to SRH recombination, bi-molecular to radiative, and tri-molecular to Auger processes [14]. Further investigation of the nature of these processes and their contribution to overall decay is required.

In the present work, we study the carrier capture processes showing that Coulomb attraction can considerably increase the SRH rate. We propose a modification of the standard SRH capture rate to account for the described effect. The proposed modification is implemented in the open-source Python library unipress (Ref. [16]), developed by the authors of this manuscript.

## 2. Results

Deep defects have their eigenstates localized in a length scale of a few Angstroms. These eigenstates serve as conversion centers from electron-hole pairs to phonons. In the SRH recombination event, the deep center is assumed to be occupied by the majority carrier. Then, the SRH process is reduced to a minority carrier capture that leads to its annihilation with the localized majority carrier. Here, we delve into this capture process.

Following Shockley, we consider first the standard SRH formulation, where it is assumed that minority carriers travel over a length l before being captured by the deep defect, with the following average constant thermal velocity:(1)$$v_{th}=\sqrt{\frac{kT}{m_{eff}}}=\sqrt{A}$$
where $m_{eff}$ is the minority carrier effective mass. The newly introduced parameter $A=v_{th}^{2}=\frac{kT}{m_{eff}}$ will be used later. The carrier capture cross-section S is estimated by assuming that the size of the deep defect is of the order of the lattice constant *a* [15]. Thus, the carrier capture cross-section is as follows: $S_{def}=\pi a^{2}$ and $S_{dis}=a$ for the capture by the point defect and by the dislocation line, respectively. In the case of an anisotropic wurtzite lattice, this estimate should be replaced by $S_{def}=\pi ac$ and $S_{dis}=c$ for point defects and dislocations, respectively. The capture condition is that the carrier encounters a single defect at the capture length l, i.e., $lN_{def}S_{def}=1$ or $lN_{dis}S_{dis}=1$, where the point defect and dislocation density are denoted by $N_{def}$ and $N_{dis}$, respectively. Note that, in general, $lS_{def}$ ($lS_{dis}$) corresponds roughly to the volume (area, respectively) swept by the carrier that traveled over distance *l* in the presence of point defects (dislocation lines); thus, $lN_{def}S_{def}$ ($lN_{dis}S_{dis}$) is the average number of encountered defects (dislocations). Therefore, the benchmark capture lengths l associated with the point defects are as follows (the data are calculated for gallium nitride (GaN)):

iFor $N_{def}=10^{17}\;cm^{-3}$,    $l=3.13\times 10^{-5}\;m$;iiFor $N_{def}=10^{19}\;cm^{-3}$,    $l=3.13\times 10^{-7}\;m$.

The benchmark capture lengths l associated with the dislocations are as follows:

iiiFor $N_{dis}=10^{7}\;cm^{-2}$,    $l=3.13\times 10^{-2}\;m$;ivFor $N_{dis}=10^{10}\;cm^{-2}$,    $l=3.13\times 10^{-5}\;m$.

For other defect densities, these values scale linearly; thus, it may be easily assessed. It is worth noting that these values are relatively large; they are much bigger than the size of quantum structures such as quantum wells. They are of the order of the typical cavity length of laser diodes. Note also that the lengths are lower for point defects that indicate the dominant role of the latter in SRH recombination.

The combination of the capture length l and the thermal velocity $v_{th}$ provides a standard estimate of the average SRH capture time:(2)$$\tau_{SRH}\left(T\right)=\frac{l}{v_{th}}=l\sqrt{\frac{m_{eff}}{kT}}$$

Note that this velocity is calculated along the path, i.e., a single component in the equipartition principle. Thus, the variation in the SRH rate from low temperature (5 K) to room temperature is $\tau_{SRH}\left(5K\right)/\tau_{SRH}\left(300K\right)\;\approx \;7.75\;$, i.e., less than one order of magnitude for the decrease from room temperature to the temperature of liquid helium. This is not a drastic decrease; nevertheless, it is frequently assumed that the SRH rate can be neglected for temperatures close to that of liquid helium [17,18,19,20].

The above comparison of the capture time at a low temperature and room temperature was based on the assumption that carriers travel with a constant thermal velocity in the crystal. However, a considerable fraction of the deep defects is charged by captured majority carriers; thus, the approaching minority carrier is attracted by a Coulomb force. As such, it is accelerated toward the defect. Additionally, the charged defect can be partially screened by the free-carrier gas, which reduces the effect. Free-carrier screening is characterized by the screening length λ.

As mentioned above, the carrier capture cross-section is relatively small, of the order of the lattice parameter *a*. Thus, only carriers that move directly toward the defect are captured, and such a process can be modeled using a one-dimensional approximation. In order to estimate the capture time, we use the energy conservation law, according to which the sum of the kinetic and potential energies is conserved:(3)$$\frac{m_{eff}v^{2}}{2}+e\;V\left(r\right)=constant$$
where $V\left(r\right)=-\frac{e\;exp\left(-r/\lambda \right)}{4\pi \varepsilon_{o}\varepsilon \;r}$ is the Coulomb screened potential for a carrier charge equal to the elementary charge, e. As the mass of the defect is at least three orders of magnitude larger than the mass of free carriers, the kinetic energy of the defect can be neglected. The velocity change on the path from infinity to a point at the distance *r* from the charged defect can be calculated as follows:(4)$$v\left(r\right)=\sqrt{\frac{kT}{m_{eff}}+\frac{e^{2}exp\left(-r/\lambda \right)}{2\pi \varepsilon_{o}\varepsilon \;rm_{eff}}}\;$$
where it was assumed that the carrier velocity at infinity is equal to the thermal velocity, i.e., $v\left(\infty \right)=v_{th}=\sqrt{\frac{kT}{m_{eff}}}$. We also assume that the distance from the center *r* in (4) is not very close to r=0, as then the Newtonian physics approximation is no longer valid for this system. The carrier velocity along the path is, therefore, position-dependent:(5)$$v\left(r\right)=\sqrt{\;A+B\left[\frac{exp\left(-r/\lambda \right)}{r}\right]}\;$$

The parameters in the above equation are as follows: $A=\frac{k_{B}T}{m_{eff}}=v_{th}^{2}$ and $B=\frac{e^{2}}{2\pi \varepsilon \varepsilon_{o}m_{eff}}$. The latter is material-dependent only and was calculated here for GaN ($\varepsilon_{GaN}=10.28)$ [21,22,23,24]. In the case of the electron capture $\left(m_{eff-GaN}=0.2\;m_{o}=1.80\times 10^{-31}\;kg\right)$, $B=2.49\times 10^{2}\;m^{3}\;s^{-2}$. The relative change in the velocity along the path could be expressed in a function of the scaled distance, $x=r/l_{Coul}$, as follows:(6a)$$\frac{v\left(x\right)}{v_{th}}=\sqrt{\;1+\left[\frac{exp\left(-\gamma x\right)}{x}\right]}$$
where the scaling parameter is the Coulomb length, defined as follows:(6b)$$l_{Coul}=\frac{B}{A}=\frac{e^{2}\;}{k_{B}T\;2\pi \varepsilon \varepsilon_{o}}\;$$

That is, the distance at which the thermal and Coulomb potential energies are equal. The Coulomb length $l_{Coul}$ is an important control parameter describing the influence of Coulomb attraction on the capture time. It is worth noting that $l_{Coul}$ is temperature dependent.

In order to grasp the value of this length, the GaN values are given as follows:

(i)For T=5K,        $l_{Coul}=3.26\times 10^{-7}m$;(ii)For T=300K,    $l_{Coul}=5.42\times 10^{-9}m$.

The larger value of $l_{Coul}$ denotes the higher influence of the Coulomb force, which is due to the temperature decrease in the thermal energy. Thus, the acceleration is more important at low temperatures. Note also that the Coulomb length is much smaller than the capture length at room temperature. At helium temperatures $\left(T\approx 5\;K\right)$ for a high point defect density, the capture and Coulomb lengths are comparable.

We introduce the parameter γ as the screening factor that determines the degree of the screening at the Coulomb length. It is equal to the Coulomb length $l_{Coul}$ divided by the screening distance λ ratio:(6c)$$\gamma =\frac{l_{Coul}\;}{\lambda }$$

The parameter γ controls the possible influence of the screening of the Coulomb potential on the capture time:

iγ ≪1.

The screening factor γ is close to zero, i.e., the screening length is much longer than the Coulomb length, i.e., $\lambda \;\gg \;l_{Coul}$; therefore, the screening does not affect the attraction potential, and the acceleration is the same as for a pure Coulomb potential.

iiγ ≫1.

In case the screening factor γ is higher than unity, screening is very effective; therefore, the capture time is determined by the thermal velocity alone, i.e., the standard SRH approach could be used. It is worth noting that if the influence of the screening is very strong, it could completely overcome the acceleration by the Coulomb force.

The magnitude of the screening distance can be determined from the Debye screening length estimate [25,26]:(7)$$\lambda =\sqrt{\frac{\varepsilon_{o}\varepsilon \;kT\;}{2e^{2}n}}\;$$
where *n* is the density of the majority carriers (electrons). For the typically encountered densities, these values are as follows (for GaN at T=300K):

i$n=10^{17}\;cm^{-3}$ $\lambda =1.21\times 10^{-8}\;m$;ii$n=10^{19}\;cm^{-3}$ $\lambda =1.21\times 10^{-9}\;m$;iii$n=10^{21}\;cm^{-3}$ $\lambda =1.21\times 10^{-10}\;m$.

The first case corresponds to a weakly doped semiconductor, and the second corresponds to a highly excited active zone in light-emitting diodes (LEDs) and laser diodes (LDs) [27,28,29,30,31,32]. Because screening lengths are generally shorter than the previously listed capture and Coulomb lengths; therefore, the case corresponding to γ ≫1 is the most frequently encountered.

These two factors have a complex influence on the carrier velocity along the path; therefore, the relative increase in the velocity as a function of the scaled distance for the different types of screened potential is plotted in Figure 1.

The data shown in Figure 1 indicate that the velocity may be accelerated in a drastically different manner. Nevertheless, several general features can be distinguished. First, the noticeable acceleration emerges at distances not larger than several Coulomb lengths. Second, the screening is extremely important; for a screening length that is close to the Coulomb length, the acceleration could be neglected at distances greater than the Coulomb length. Finally, at very close distances, the acceleration may increase the carrier velocity by one order of magnitude.

These data indicate that the acceleration is important, especially relatively close to the charged center and in the absence of effective screening. It is expected, however, that the carrier may travel longer distances. This leads to a reduction in the capture time, which cannot be estimated from the magnitude of the velocity directly. It also has to be assumed that the carrier has a thermal velocity at the beginning of the path, i.e., for r=l. Thus, the velocity is changed to the following:(8)$$\tilde{v}\left(r\right)=\sqrt{A+B\left[\frac{exp\left(-r/\lambda \right)}{\;r}-\frac{exp\left(-l/\lambda \right)}{\;l}\right]}\;$$

Therefore, the screened Coulomb attraction-dependent capture time $\tau_{S-Coul}$ has to be calculated as the time of the arrival of the carrier from the distance *l* to the recombination center:(9)$$\tau_{S-Coul}=\int_{0}^{l}\frac{dr}{\tilde{v}\left(r\right)}=\int_{0}^{l}\frac{dr}{\sqrt{A+B\left[\frac{exp\left(-r/\lambda \right)}{\;r}-\frac{exp\left(-l/\lambda \right)}{\;l}\right]}}\;$$

Since $l_{Coul}=B/A$ and $\gamma =l_{Coul}/\lambda$, Expression (9) may be simplified by substituting y=r/l, followed by $\vartheta =l/l_{Coul}\;$. Then, the capture time (9) may be expressed using the SRH capture time, $\tau_{SRH}=\vartheta \frac{B}{A^{3/2}}$, and the capture length/Coulomb length ratio ϑ  as follows:(10)$$\tau_{S-Coul}\left(\vartheta \right)=\tau_{SRH}\int_{0}^{1}\frac{dy}{\sqrt{1-\frac{exp\left(-\gamma \vartheta \right)}{\vartheta }+\frac{exp\left(-\gamma \vartheta y\right)}{y\vartheta }}}$$
where we neglected the size of the defect, as it is small compared with the capture length, and additionally, the carrier velocity at the capture is very high due to acceleration, and the integral was extended to zero, as its contribution is small due to the very high velocity in this region.

First, we considered the acceleration caused by the influence of the Coulomb potential only (γ=0) on the capture time, (i.e., without screening—$\tau_{Coul}$) (10):(11)$$\tau_{Coul}=\tau_{SRH}\int_{0}^{1}\frac{dy}{\sqrt{1+\frac{1}{\vartheta }\left(\frac{1}{y}-1\right)}}$$

This integral could be evaluated analytically. For y>0 and constants a,b>0, we have $\int \frac{dy}{\sqrt{a+\frac{b}{y}}}=\frac{\sqrt{ay^{2}+b}}{a}-\frac{b\;asinh\left(\sqrt{\frac{ay}{b}}\right)}{a\sqrt{a}}$, and if, additionally, ay≤b, then $\int \frac{dy}{\sqrt{-a+\frac{b}{y}}}=\frac{y\sqrt{y}}{\sqrt{b-ay}}-\frac{b\sqrt{y}}{a\sqrt{b-ay}}+\frac{b\;asin\left(\sqrt{\frac{ay}{b}}\right)}{a\sqrt{a}}$. If we substitute $a=1-\frac{1}{\vartheta },\;b=\frac{1}{\vartheta }$ for ϑ>1 in the first integral and $a=\frac{1}{\vartheta }-1,\;b=\frac{1}{\vartheta }$ for 0<ϑ<1 in the second integral, satisfying ay≤b for 0≤y≤1, and for ϑ=1 using $\int \frac{dy}{\sqrt{\frac{1}{y}}}=\frac{2y\sqrt{y}}{3}$, and then integrate over $y\in \left[0,1\right]$, Expression (11) gives the following:(12)$$\tau_{Coul}\left(\vartheta \right)=\tau_{SRH}\frac{\vartheta }{\vartheta -1}\left[1-\frac{1}{\sqrt{\vartheta }\sqrt{\vartheta -1}}\;asinh\left(\sqrt{\vartheta -1}\right)\right]\;\;\;\vartheta >1\;\;\;\;\;\;\;\;\;\;\;\;\frac{2}{3}\tau_{SRH}\;\;\;\;\;\;\;\;\;\;\;\;\vartheta =1\;\;\;\;\;\;\;\;\;\;\;\;\;\tau_{SRH}\frac{\sqrt{\vartheta }asin\left(\sqrt{1-\vartheta }\right)-\vartheta \sqrt{1-\vartheta }\;}{{\left(1-\vartheta \right)}^{3/2}}\;\;\;\;\;\;0<\vartheta <1\;$$

This dependence could be converted into the following:(13)$$\;\tau_{Coul}\left(\vartheta \right)=\tau_{SRH}\frac{\vartheta }{\vartheta -1}\left[1-\frac{1}{\sqrt{\vartheta }\sqrt{\vartheta -1}}\;ln\left(\sqrt{\vartheta -1}+\sqrt{\vartheta }\right)\right]\;\vartheta >1\;\;\;\;\;\;\;\;\;\;\;\frac{2}{3}\tau_{SRH}\;\;\;\;\;\;\;\;\;\;\;\vartheta =1\;\tau_{SRH}\frac{\sqrt{\vartheta }asin\left(\sqrt{1-\vartheta }\right)-\vartheta \sqrt{1-\vartheta }\;}{{\left(1-\vartheta \right)}^{3/2}}\;\;\;\;\;\;\;\;\;\;\;\;\;0<\vartheta <1$$

This result can be approximated as a function of the capture and Coulomb lengths ratio ϑ for two opposite regimes:

iWeak attraction $l\;\gg \;l_{Coul}$ $\left(\vartheta \;\gg 1\right)$



(14a)
$$\tau_{Coul}≅\tau_{SRH}\left\{1-\frac{ln\left(4\vartheta \right)}{2\vartheta }\right\}\;$$



This relation recovers the SRH capture time for very weak Coulomb attraction $\left(\;l_{Coul}/l\;\to 0\right)$.

iiStrong attraction $\;l_{Coul}\;\gg l$ $\left(\vartheta \;\ll 1\right)$



(14b)
$$\tau_{Coul}≅\frac{\pi \tau_{SRH}}{2}\sqrt{\vartheta }\;$$



This reflects the reduction in the capture time for strong Coulomb attraction by the charged recombination center.

A more complex picture emerges in the case of the screened Coulomb potential. The screened Coulomb capture time may be expressed as follows:(15)$$\tau_{S-Coul}=\tau_{SRH}\int_{0}^{1}\frac{dy}{\sqrt{1-\frac{exp\left(-\gamma \vartheta \right)}{\vartheta }+\frac{exp\left(-\gamma \vartheta y\right)}{y\vartheta }}}$$

The integral in Equation (15) cannot be evaluated analytically; therefore, numerical integration was employed to obtain the capture time dependence on ϑ for various values of γ. The data obtained are plotted in Figure 2.

These results indicate that the Coulomb attraction affects the capture time over large distances. The acceleration leads to a shortening of the time, even for a capture length as high as 10^2^ Coulomb lengths. This is in agreement with the long-range character of electrostatic interactions. Nevertheless, the effect is not large, and the capture time is shortened by 20% for 10 Coulomb lengths. On the other hand, for a close distance, this reduction is more substantial: for a capture length equal to the Coulomb length, the reduction is by one-third. A drastic reduction is observed for very close distances, below $0.5\;l_{Coul}$. In contrast to the long-range behavior of the Coulomb force, screening is governed by the exponential law, i.e., it is short-ranged. This is reflected by the screened attraction data.

The wide and effective application of this formulation requires an effective way of calculating the capture time that is comparable to the standard SRH estimate. In order to avoid a somewhat cumbersome integration, an approximate expression was obtained:(16)$$\tau_{S-Coul}\left(\vartheta \right)≅\tau_{SRH}{\left(\frac{\vartheta^{P1}}{\vartheta^{P1}+P2}\right)}^{P3}$$
where the approximation parameters are given as functions of γ:(17)$$P1\left(\gamma \right)=0.949+0.049\;log\left(\gamma \right)$$
(18)$$P2\left(\gamma \right)=0.861-0.169\;log\left(\gamma \right)$$
(19)$$P3\left(\gamma \right)=0.475+0.005327\;log\left(\gamma \right)+0.001699\;{\left[log\left(\gamma \right)\right]}^{2}$$

These expressions were verified to be used in the following range: $\gamma \;\in \;\left[0.01,\;100.0\right]$. In total, Equations (16) and (17) recover the accelerated time in the range $\vartheta \in \;\left[0.01,\;100\right]$, with a precision of 35 percent.

For the ease of use of the provided expressions, both Approximation (16) and Formula (10), calculated by a numerical quadrature, are integrated into open-source Python library unipress [16].

The dependencies presented in Figure 2 and the data quoted earlier could be used to assess the relative importance of the reduction in the capture time. As is shown, in the pure Coulomb case without screening effects (i.e., γ=0), the reduction is not limited to the Coulomb length, i.e., to $\vartheta =l/l_{Coul}\approx 1$. For this case, the reduction is of the order of 40% (see Figure 2b). Nevertheless, it extends longer, $\vartheta =l/l_{Coul}\approx 10$, which is still 20%, while in the case of $\vartheta =l/l_{Coul}\approx 100$, it is about 5% (see Figure 2a). On the other hand, for a relatively short capture length, such as $\vartheta =l/l_{Coul}\approx 0.1$, it is one order of magnitude smaller (see Figure 3). Note that, for T=5 K, we obtained $l_{Coul}=3.26\times 10^{-7}\;m$; therefore, such an extreme case can be achieved for high point defect densities, of about $N_{def}\approx \;10^{19}\;\;{\mathrm{cm}}^{-3}$. For this defect density, the obtained SRH time is $\tau_{SRH}\;\approx 1.7\times 10^{-11}\;s$. The obtained pure Coulomb time is $\tau_{Coul}\;\approx 2.3\times 10^{-12}\;s$, which is considerably shorter. The screened length of the relatively low free electron density $n=10^{15}{\;\mathrm{cm}}^{-3}$ is λ=10  nm, which results in the intermediate time $\tau_{S-Coul}\;\approx 1.4\times 10^{-12}\;s$. Thus, the acceleration effects could be important for low-temperature PL investigations, where SRH recombination should not be neglected. For higher temperatures, i.e., T=300 K, this acceleration-related effect is considerably smaller: $\tau_{SRH}\;\approx 2.0\times 10^{-12}\;s$, $\tau_{Coul}\;\approx 1.5\times 10^{-12}\;s$, and $\tau_{S-Coul}\;\approx 1.8\times 10^{-12}\;s$ for SRH, pure Coulomb, and screened Coulomb, respectively.

These effects should be investigated in more detail, which we plan to do in further studies, including the determination of the SRH recombination rate in the function of the point defect and dislocation densities, using newly developed methods of the analysis of time-resolved photoluminescence (TRPL) signal decay, as described in Ref [14]. The planned research will include the role of the defects in the radiative recombination rates.

## 3. Summary

It is shown that the electrostatic attraction of the minority carriers by charged deep defects may reduce the carrier capture time, significantly increasing the SRH recombination rate. The effect is related to the rate-limiting step, i.e., the carrier capture in linear nonradiative (i.e., SRH) recombination. The reduction in the capture time is expressed as a function of the two control parameters: the capture length/Coulomb length ratio and the screening length/Coulomb length ratio. The Coulomb length is defined as the length at which the Coulomb energy is equal to the kinetic motion thermal energy, i.e., it is temperature dependent. It was demonstrated recently that, in the typical semiconductor device or semiconductor medium, SRH recombination cannot be neglected at low temperatures [33,34].

Effective screening by mobile carrier density could reduce the acceleration effect, leading to an SRH decrease. This effect may be responsible for the decrease in LED efficiency at high excitations, known as “droop effect”.

The described effect of excited carriers’ acceleration due to the electrostatic attraction of the minority carriers by charged deep defects could enhance SRH recombination rates above the thermally imposed limits, showing the possibility of the nonradiative recombination increase, which was not explained by the standard SRH formulation.

## Figures and Tables

**Figure 1 materials-17-04581-f001:**
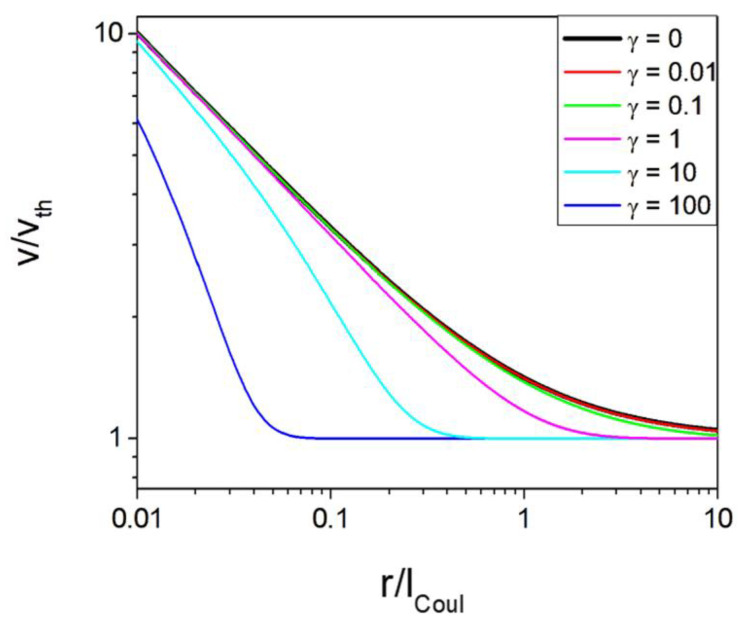
The relative increase in the velocity of the carriers as a function of the scaled distance *x* from the charged defect, calculated for several values of the screening fraction γ. The line (red) for γ=0 corresponds to the Coulomb potential and is essentially identical to γ=0.01.

**Figure 2 materials-17-04581-f002:**
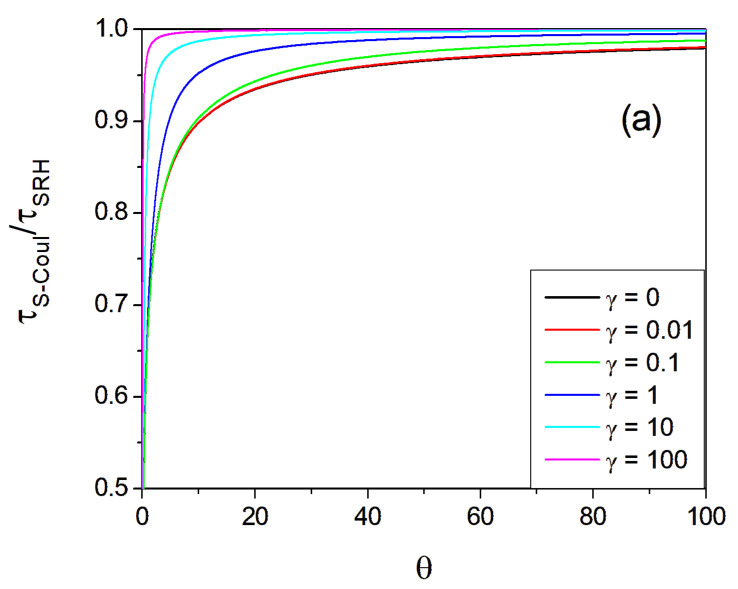
The relative capture time (capture/SRH time ratio, i.e., $\tau_{S-Coul}/\tau_{SRH}$ ) in a function of the scaled capture length (capture/Coulomb length ratio, i.e., $\vartheta =l/l_{Coul}$) for various values of the screening factor $\gamma =\frac{l_{Coul}\;}{\lambda }$. γ=0 corresponds to λ → ∞, i.e., pure Coulomb attraction: (**a**) a wide range of ϑ values, (**b**) a magnified portion of small ϑ. The line (red) for γ=0 corresponds to the Coulomb potential and is essentially identical to γ=0.01.

**Figure 3 materials-17-04581-f003:**
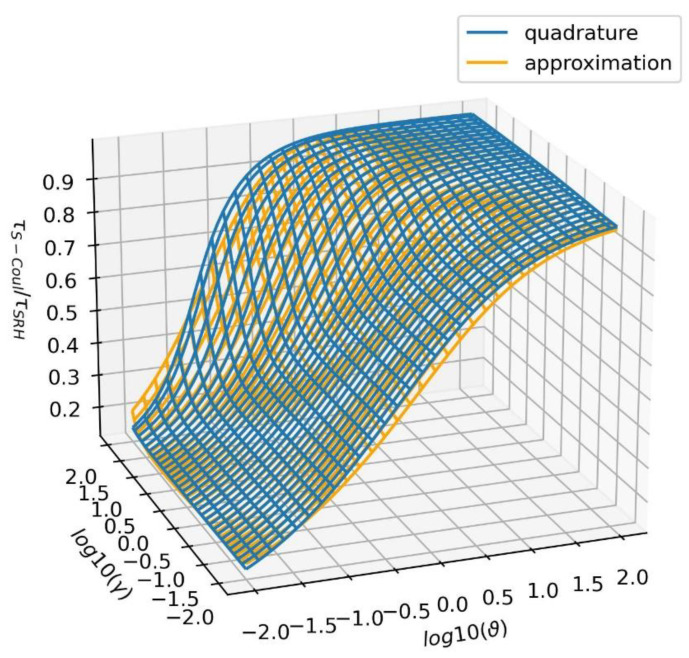
The relative capture time ($\tau_{S-Coul}/\tau_{SRH}$) calculated by Formula (10) by a numerical quadrature and by Approximation (16).

## Data Availability

The raw data supporting the conclusions of this article will be made available by the authors on request.

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
