# Peer review of "Coulomb Contribution to Shockley–Read–Hall Recombination"

_materials, 2024, doi:10.3390/ma17184581_

Round 1

Reviewer 1 Report

Comments and Suggestions for Authors

In this work, the authors proposed a view that in a typical semiconductor device or semiconductor medium, SRH recombination rate at low temperatures is much higher and it cannot be neglected, which was demonstrated by a theoretical analysis. This insight, supported by theoretical analysis, is valuable for understanding carrier recombination dynamics. The work is of potential interest for publishing after necessary revisions. Here are some suggestions that the authors could pick up on:

1. In introduction, the authors write: “SRH is one of the most important detrimental processes for optoelectronic devices, having a negative impact on the performance of all semiconductor light emitters, including laser diodes (LDs), light emitting diodes (LEDs), superluminescent light emitting diodes (SLEDs), and electroluminescent lamps. [1, 2]…The recent advent of nitride optoelectronic confirmed the important role of SRH recombination…It is widely recognized that, in InGaN LEDs, carrier localization in In-rich areas prevents migration to dislocation lines, which usually serve as effective nonradiative SRH recombination centers in semiconductors. [7, 8, 9] Such a positive insulation effect is not observed in AlGaN-based devices, which leads to a lower emission efficiency in comparison to their InGaN-based counterparts. [11]” The general reference list in the introduction seems a bit thin, considering the evolution in the field within the recent years. To give the readers a much broader view, recent developments concerning on SRH, InGaN LEDs and AlGaN-based devices, such as Laser & Photonics Reviews 2024, 18, 2300464, etc. should be added, so that the readers can be clear about the state-of-the-art of this topic.

2. In line 102, the authors write: “Thus, the carrier capture cross-section is: 𝑆𝑑𝑒𝑓 = 𝜋𝑎2 and 𝑆𝑑𝑖𝑠 = 𝑎 for the capture by the point defect and by the dislocation line, respectively. ” Given the hexagonal wurtzite structure of III-nitride materials, why is the cross-section of the point defect assumed to be circular rather than hexagonal?

3. In line 183, the authors list the typical majority carrier densities for GaN at 300K. However, the electron density is either not equal to or significantly higher than the hole density due to the considerable difference in donor and acceptor activation energies. How does this factor influence the calculation of screening distance?

4. In line 300, the authors write: “Effective screening by mobile carrier density could reduce the acceleration effect, leading to SRH decrease. The effect may be responsible for the decrease of LED efficiency at high excitation known as ‘droop effect’.” How does Auger recombination impact this at high excitation levels?

5. In an overview of this paper, the authors assert that the SRH recombination rate at cryogenic temperatures cannot be neglected. However, how do you explain the common experimental observation that PL/EL integral intensity at room temperature is lower than at cryogenic temperatures? Please relate this to your proposed theory.

Author Response

Response to Reviewer 1 Comments

1. Summary

Thank you very much for taking the time to review this manuscript. Please find the detailed responses below and the corresponding revisions/corrections highlighted by blue color /in track changes in the pdf file. 

2. Questions for General Evaluation

Reviewer’s Evaluation

Response and Revisions

Does the introduction provide sufficient background and include all relevant references?

Yes/Can be improved/Must be improved/Not applicable

Are all the cited references relevant to the research?

Yes/Can be improved/Must be improved/Not applicable

We have added Reference as suggested by the Referee. Please renumber the references staring from Ref 12.

Is the research design appropriate?

Yes/Can be improved/Must be improved/Not applicable

Are the methods adequately described?

Yes/Can be improved/Must be improved/Not applicable

Are the results clearly presented?

Yes/Can be improved/Must be improved/Not applicable

Are the conclusions supported by the results?

Yes/Can be improved/Must be improved/Not applicable

3. Point-by-point response to Comments and Suggestions for Authors

Comments 1: In introduction, the authors write: “SRH is one of the most important detrimental processes for optoelectronic devices, having a negative impact on the performance of all semiconductor light emitters, including laser diodes (LDs), light emitting diodes (LEDs), superluminescent light emitting diodes (SLEDs), and electroluminescent lamps. [1, 2]…The recent advent of nitride optoelectronic confirmed the important role of SRH recombination…It is widely recognized that, in InGaN LEDs, carrier localization in In-rich areas prevents migration to dislocation lines, which usually serve as effective nonradiative SRH recombination centers in semiconductors. [7, 8, 9] Such a positive insulation effect is not observed in AlGaN-based devices, which leads to a lower emission efficiency in comparison to their InGaN-based counterparts. [11]” The general reference list in the introduction seems a bit thin, considering the evolution in the field within the recent years. To give the readers a much broader view, recent developments concerning on SRH, InGaN LEDs and AlGaN-based devices, such as Laser & Photonics Reviews 2024, 18, 2300464, etc. should be added, so that the readers can be clear about the state-of-the-art of this topic.

Response 1: We fully agree with the Referee that the subject is huge and growing quickly. Therefore we were not able to cite all references as it is not possible. We have selected some and most likely we overlooked some important positions for which we apologize. Therefore, we thank the Referee for their suggestion with appropriate comments. We have added the reference pointed out by the Referee. This is fairly recent publication. Hopefully, this will improve the generally picture of the field considerably.

Comments 2: In line 102, the authors write: “Thus, the carrier capture cross-section is: ???? = ??2 and ???? = ? for the capture by the point defect and by the dislocation line, respectively. ” Given the hexagonal wurtzite structure of III-nitride materials, why is the cross-section of the point defect assumed to be circular rather than hexagonal?

Response 2: Yes we agree that these crystal are strongly anisotropic, so it is natural that the size of the deep defect may be different along different directions, i.e. in the plane perpendicular and along c-axis. The cross-section listed in the manuscript is an estimate, which is fairly general, based on general physics of such defects. Generally, these defects that are much bigger. They are shallow defects, not active in SRH recombination. Therefore, the size of the unit cell is a good estimate for the size of the SRH active deep defect. Nevertheless, it is possible that this estimate in case of point defect  should be replaced by  and in case of dislocations this estimate is  should be replaced by . This is an increase by the factor smaller than 2 hence it is applicable for both lattices. The order of magnitude of the defect size remains the same. We have corrected the text to account the anisotropy.

Comments 3: In line 183, the authors list the typical majority carrier densities for GaN at 300K. However, the electron density is either not equal to or significantly higher than the hole density due to the considerable difference in donor and acceptor activation energies. How does this factor influence the calculation of screening distance?

Response 3: This is our omission, we referred to electron density naturally. We thank the Referee for the this remark. We corrected the text accordingly. Moreover, we would like to mention that the presented theory is also applicable for holes as a majority carrier in analogous way. Impact of minority carriers on screening is assumed to be negligible relative to majority carriers.

Comments 4: In line 300, the authors write: “Effective screening by mobile carrier density could reduce the acceleration effect, leading to SRH decrease. The effect may be responsible for the decrease of LED efficiency at high excitation known as ‘droop effect’.” How does Auger recombination impact this at high excitation levels?

Response 4: We fully agree that the Auger effect may contribute to the well-known droop effect. However at this point we cannot say anything interesting on this effect because we do not have any new information.

Comments 5: . In an overview of this paper, the authors assert that the SRH recombination rate at cryogenic temperatures cannot be neglected. However, how do you explain the common experimental observation that PL/EL integral intensity at room temperature is lower than at cryogenic temperatures? Please relate this to your proposed theory.

Response 5: The results in our paper is that the SRH is lower at cryogenic temperatures but it is not negligible. Thus we agree that the SRH competition to radiative recombination is lower at low temperatures, and that is why PL/EL intensity is higher at very low temperatures. We agree with that fully. The difference is that SRH reduction within new model in not so large as it was in classical model.

4. Response to Comments on the Quality of English Language

Point 1:

Response 1:    None

5. Additional clarifications

1.      In the pdf file the new text is marked by blue color.

2.      In submittion we have not added Pawel Strak as an author. He is in the manuscript so we added him as a correction.

Reviewer 2 Report

Comments and Suggestions for Authors

Please indicate more in detail where do come from the values from line 111 to 115 on the page 3.

We cannot observe the curve for gamma = 0 in Fig.1 and Fig. 2. As these are extreme cases, indicate the shape maybe?

Aand what happens in the intermediate temperatures?

Author Response

Response to Reviewer 2 Comments

1. Summary

Thank you very much for taking the time to review this manuscript. Please find the detailed responses below and the corresponding revisions/corrections highlighted by blue color /in track changes in the pdf file. 

2. Questions for General Evaluation

Reviewer’s Evaluation

Response and Revisions

Does the introduction provide sufficient background and include all relevant references?

Yes/Can be improved/Must be improved/Not applicable

Are all the cited references relevant to the research?

Yes/Can be improved/Must be improved/Not applicable

Is the research design appropriate?

Yes/Can be improved/Must be improved/Not applicable

Are the methods adequately described?

Yes/Can be improved/Must be improved/Not applicable

Are the results clearly presented?

Yes/Can be improved/Must be improved/Not applicable

Are the conclusions supported by the results?

Yes/Can be improved/Must be improved/Not applicable

3. Point-by-point response to Comments and Suggestions for Authors

Comments 1: We cannot observe the curve for gamma = 0 in Fig.1 and Fig. 2. As these are extreme cases, indicate the shape maybe?

Response 1: The curve  correspond to pure Coulomb case (black) are overlapped by the case  (red). They are identical, so it is overlapped. We have added phrase in the captions. We thank for pointing us the problem.

Comments 2: Aand what happens in the intermediate temperatures?

Response 2: For intermediate temperatures these formulae apply, the reduction is less pronounced.

4. Response to Comments on the Quality of English Language

Point 1:

Response 1:    None

5. Additional clarifications

1.      In the pdf file the new text is marked by blue color.

2.      In submittion we have not added Pawel Strak as an author. He is in the manuscript so we added him as a correction.

Reviewer 3 Report

Comments and Suggestions for Authors

Konrad Sakowski et al. revisit in their manuscript entitled “Coulomb Contribution to Shockley-Read-Hall Recombination” a seminal model for an important nonradiative relaxation mechanism of electron-hole pairs in semiconductors. As pointed out in the introduction of the manuscript, this so-called Shockley-Read-Hall (SRH) recombination at deep majority-carrier traps can be responsible for low efficiency of optical emitters and is particularly important in many Nitride semiconductors due to the high trap densities in these materials. The paper hence deals with a fundamental mechanism observed in many materials of technological importance and thus is of high interest to a broad readership.

In their report, the authors point out that unlike in the standard description of the SRH mechanism the attractive interaction between the trapped charge and the minority carriers may not be neglected in some cases. They develop an easy-to-follow and convincing model carefully pointing out the conditions at which the Coulomb interaction leads to an increase of the RSH-recombination rate. It turns out that, relative to the standard SRH model, the rates obtained in the new model can be enhanced by roughly an order of magnitude in the limiting case of low temperature, high trap density and low majority carrier density. The reader looks forward to experimental studies that the authors plan in order to confirm the outcome of the model.   

The paper is well written, convincing and reports results of high interest. Hence, I can recommend publication of the work in the Materials journal. 

Before publication the authors might want to consider the following comments in order to even further improve readability of the article.  

-          Line 36: Replace “remains” by “remained”.  

-          Lines 70-71: Replace “as then only the radiative recombination channel is significant.” by “as it is generally believed that then only the radiative recombination channel would be significant.”

-          Line 72: Replace “not sufficiently high.” by “sufficiently low.”.

-          Lines 111-115: Please mention which material (lattice parameter) is assumed for the calculation of the quoted values.

-          Line 200: Replace “showed” by “shown”.

-          Lines 296-300: Skip the repetition of the sentence.

Comments on the Quality of English Language

good

Author Response

Response to Reviewer 3 Comments

1. Summary

Thank you very much for taking the time to review this manuscript. Please find the detailed responses below and the corresponding revisions/corrections highlighted by blue color /in track changes in the pdf file. 

2. Questions for General Evaluation

Reviewer’s Evaluation

Response and Revisions

Does the introduction provide sufficient background and include all relevant references?

Yes/Can be improved/Must be improved/Not applicable

Are all the cited references relevant to the research?

Yes/Can be improved/Must be improved/Not applicable

.

Is the research design appropriate?

Yes/Can be improved/Must be improved/Not applicable

Are the methods adequately described?

Yes/Can be improved/Must be improved/Not applicable

Are the results clearly presented?

Yes/Can be improved/Must be improved/Not applicable

Are the conclusions supported by the results?

Yes/Can be improved/Must be improved/Not applicable

3. Point-by-point response to Comments and Suggestions for Authors

Comments 1: Line 36: Replace “remains” by “remained

Response 1: Corrected. Thank you.

Comments 2: Lines 70-71: Replace “as then only the radiative

recombination channel is significant.” by “as it is generally believed

that then only the radiative recombination channel would be

significant.”

Response 2: Corrected. Thank you

Comments 3: Line 72: Replace “not sufficiently high.” by “sufficiently low.”.

Response 3: Corrected. Thank you

Comments 4: Line 72: Replace “not sufficiently high.” by “sufficiently low.”.

Response 4: Corrected. Thank you

Comments 5: Lines 111-115: Please mention which material (lattice

parameter) is assumed for the calculation of the quoted values

Response 5: That was Gallium nitride. Introduced in the text.

Comments 5: Line 200: Replace “showed” by “shown”.

Response 5: Replaced. Thank you

Comments 6: Lines 296-300: Skip the repetition of the sentence.

Response 56: Removed. Thank you

4. Response to Comments on the Quality of English Language

Point 1:

Response 1:    None

5. Additional clarifications

1.      In the pdf file the new text is marked by blue color.

2.      In submission we have not added Pawel Strak as an author by omission. He is in the manuscript so we added him as a correction.

Reviewer 4 Report

Comments and Suggestions for Authors

The article "Coulomb Contribution to Shockley-Read-Hall Recombination" considers important calculations for understanding nonradiative recombination.

However, applicability of the results is complicated due to dimensionless parameters. The authors should calculate nonradiative lifetime dependence on carrier, defect density and temperature to show its values eg. in nanoseconds to compare with room temperature data to extrapolate that to low temperatures and then discuss with available data. It is suspected if lifetime at RT is only few nanoseconds, then at low temperatures it will contribute significantly to PL efficiency reduction. 

Author Response

Response to Reviewer 4 Comments

1. Summary

Thank you very much for taking the time to review this manuscript. Please find the detailed responses below and the corresponding revisions/corrections highlighted by blue color /in track changes in the pdf file. 

2. Questions for General Evaluation

Reviewer’s Evaluation

Response and Revisions

Does the introduction provide sufficient background and include all relevant references?

Yes/Can be improved/Must be improved/Not applicable

Are all the cited references relevant to the research?

Yes/Can be improved/Must be improved/Not applicable

.

Is the research design appropriate?

Yes/Can be improved/Must be improved/Not applicable

Are the methods adequately described?

Yes/Can be improved/Must be improved/Not applicable

Are the results clearly presented?

Yes/Can be improved/Must be improved/Not applicable

Are the conclusions supported by the results?

Yes/Can be improved/Must be improved/Not applicable

3. Point-by-point response to Comments and Suggestions for Authors

Comments 1: However, applicability of the results is complicated due to dimensionless parameters. The authors should calculate nonradiative lifetime dependence on carrier, defect density and temperature to show its values eg. in nanoseconds to compare with room temperature data to extrapolate that to low temperatures and then discuss with available data. It is suspected if lifetime at RT is only few nanoseconds, then at low temperatures it will contribute significantly to PL efficiency reduction. 

Response 1: We agree that this is a problem so we thank the Referee for this point. Accordingly we have provided numerical estimate for several cases in case of gallium nitride, grown by MOVPE of the electron carrier density is  and high pressure grown GaN crystals of the higher value of electron density, i.e.  at cryogenic and room temperatures, i.e.  and

4. Response to Comments on the Quality of English Language

Point 1:

Response 1:    None

5. Additional clarifications

1.      In the pdf file the new text is marked by blue color.

2.      In submittion we have not added Pawel Strak as an author. He is in the manuscript so we added him as a correction.

Round 2

Reviewer 4 Report

Comments and Suggestions for Authors

The revised version addresses all comments and can be published as is.